# Quality of Oil Pressed from Hemp Seed Varieties: ‘Earlina 8FC’, ‘Secuieni Jubileu’ and ‘Finola’

**DOI:** 10.3390/molecules27103171

**Published:** 2022-05-16

**Authors:** Wojciech Golimowski, Mirosława Teleszko, Damian Marcinkowski, Dominik Kmiecik, Anna Grygier, Andrzej Kwaśnica

**Affiliations:** 1Department of Agroengineering and Quality Analysis, Faculty of Engineering and Economics, Wroclaw University of Economics and Business, Komandorska 180/120, 53-345 Wrocław, Poland; damian.marcinkowski@ue.wroc.pl; 2Department of Food Technology and Nutrition, Faculty of Production Engineering, Wroclaw University of Economics and Business, Komandorska 118/120, 53-345 Wrocław, Poland; miroslawa.teleszko@ue.wroc.pl; 3Department of Food Technology of Plant Origin, Faculty of Food Science and Nutrition, Poznań University of Life Sciences, Wojska Polskiego 31, 60-624 Poznan, Poland; dominik.kmiecik@up.poznan.pl (D.K.); anna.grygier@up.poznan.pl (A.G.); 4Department of Food Chemistry and Biocatalysis, Wroclaw University of Life Sciences, C.K. Norwida Street 25, 50-375 Wrocław, Poland; andrzej.kwasnica@upwr.edu.pl

**Keywords:** *Cannabis sativa* L., sterols, hemp oil, ‘Finola’, ‘Earlina8FC’, Secuieni Jubileu

## Abstract

In the last decade, the demand for edible niche oils has increased. Therefore, the aim of this study was to characterize the seeds hemp (*Cannabis sativa* L.) varieties: ‘Finola’ (FIN-314)’, ‘Earlina 8FC’, and ‘Secuieni Jubileu’, and cold and hot pressed oils were prepared from each seed. The seeds were examined for moisture content, granulometric distribution, bulk density, and fat content. Seeds were pressed without and with preconditioning (60 °C), and oil yield and pressing time were recorded. The oil was filtered through cellulose membranes. Oil–water content, oil color, fatty acid profile, and sterol content were studied. From the study conducted, there are significant differences in the parameters of oil recovery and its quality compared to ‘Finola’ seed oil, which is widely reported in the literature. ‘Finola’ oil yield was the lowest, with an average of 79% compared to ‘Earlina’ (82%) and ‘S. Jubileu’ (84%). All oil samples contained a comparable amount of sterols, with campesterol (0.32 mg/g), β-sitosterol (1.3 mg/g) and Δ5-avenasterol (0.15 mg/g) predominating. From the organoleptic evaluation, it was evident that both varieties hemp oils and marc (‘Earlina’ and ‘S. Jubileu’) were not bitter like the “Finola” oil and marc. More detailed studies in this direction have to be undertaken.

## 1. Introduction

Hemp (*Cannabis sativa* L.) is characterized by interesting properties and is used in food, medical and clothing industries [1,2], and due to psychoactive ingredients (D9-THC), it is also used for therapeutic purposes [3]. Depending on the variety, hemp essential oils are rich in phytochemicals, especially terpenoids [4]. Hemp is mainly grown in Europe, Asia, and North America [5]. Hemp fiber is used for clothing, paper, building insulation, and composite materials [6,7]. Hemp seeds are a rich source of vitamins A, C, and E, minerals, and β-carotene and contain approximately 30% protein, 25% starch, and 30% oil [8]. They are pressed to obtain oil, and the meal is processed into flour [9,10,11]. There is also a great interest in the positive health-promoting effects of non-psychoactive cannabinoids, i.e., cannabidiol (CBD) and its precursor cannabidiolic acid (CDBA) [12]. Hemp oil is rich in nutrients and health promoting substances, i.e., vitamins, mineral salts, amino acids, and phytosterols [8]. This oil contains 80% polyunsaturated fatty acids (PUFA), especially α-linolenic acid (18:3; n-3) and linoleic acid (18:2; n-6). The ratio of n6 to n3 acids in hemp oil is 3:1, which is beneficial for human health [13,14]. Hemp oil can also be used to produce biodiesel [15,16]. Given the ever-increasing demand for vegetable oils and consumer awareness of their nutritional and health-promoting role in the human diet, the composition of the unsaponifiable fraction of hemp oil is still not well studied. This fraction constitutes about 1.5–2% of the oil and consists mainly of plant sterols, also called phytosterols [17]. Phytosterols are by-products of the isoprenoid biosynthesis pathway from acetyl-coenzyme-A [18]. They are characterized by anticancer [18,19,20], anti-inflammatory [21], and antidiabetic [22] properties. Studies have shown that consumption of 2 g of plant sterols per day can reduce the risk of cardiovascular disease by up to 25% [18]. The profile and total content of phytosterol in vegetable oils depend on many factors: agronomic and climatic conditions, pressing and extraction methods, and storage conditions [23]. The phytosterol profile can be used to confirm authenticity or to detect adulteration in the composition of selected vegetable oils [24].

The sterol content in hemp oil can vary from 2.2 to 6.7 g/kg of oil. Their content may depend on many factors: plant variety, place of origin, cultivation conditions, or pressing conditions [25]. The main sterols in hemp oil are β-sitosterol and campesterol. On average, they account for 81 to 85% of all phytosterols. In addition, we find in the unsaponifiable fraction other plant sterols and stanols such as stigmasterol, stigmastanol, campestanol, sitostanol, Δ5-, and Δ7-avenastreol and clerosterol [17]. The dominant sterol fraction in hemp oil is the Δ5-sterol components. The sterols with the Δ7 configuration are the secondary fraction. The content of sterols with the Δ7 configuration is only 2% of all plant sterols in hemp oil [25].

‘Finola’ is the official name of a Finnish hemp variety that was approved as a European Union variety in 2004. Compared to other hemp varieties, ‘Finola’ is relatively short (maximum height is 1.5 m) and is tolerant to drought and low temperatures at all growth stages [7]. As shown in previous studies, ‘Finola’ seeds contain about 30% oil and 25% protein. The disadvantage of this variety is the small grain size, which is half the size of its most popular varieties [1].

Hemp is a valuable source of protein and vegetable oil. About 600 varieties of this plant have been identified, of which there is not much information in the literature regarding oil production. The aim of this study was to compare two little known varieties ‘Earlina 8FC’ and ‘S. Jubileu’ with the well-known cultivar ‘Finola’ in terms of vegetable oil production. The research covered the entire oil production process, from its collection to the determination of the quality of the obtained oil. The effect of plant variety and seed pressing parameters on the quality and quantity of obtained oil was studied.

## 2. Results and Discussion

### 2.1. The Efficiency of the Pressing Process

Process parameters in oilseed pressing have a significant effect on oil yield and quality [5]. Table 1 shows the results of single-stage oil pressing of seeds at different temperatures by pressing them with different technical parameters of the press. The efficiency of the press was calculated from the ratio between the weight of seeds (M_o_) and pressing time (Δt) using the formula:(1)$$P=\frac{M_{s}}{∆t}\;\left[\frac{\mathrm{kg}}{h}\right]$$

The process efficiency was calculated based on the weight of fat in the seed (M_s_∗O) and the weight of the oil after pressing (M_o_) relative to the seeds according to the formula:
(2)$$E=\frac{M_{o}}{M_{s}*O}\times 100\%\;\left[\%\right]$$

The alternative independent variables were seed varieties, temperature of pressed seed, and press shaft speed. Mean press head temperature, press performance (P), and efficiency of the pressing process (E) were considered as dependent variables. ANOVA statistical analyses showed that the independent variables had a significant effect on the dependent variables. The correlation analyses (Cc—Correlation coefficient) show that seed temperature has a strong effect on process temperature (Cc = 0.9), press head capacity (Cc = 0.68), and process efficiency (Cc = 0.53). Increasing the in shaft speed decreased the pressing temperature (Cc = −0.24), and process efficiency (Cc = −0.71), while it significantly affected the increase in performance (Cc = 0.67). The yield had no significant effect on pressing efficiency. The pressed oil was filtered using a plate filter. Oil samples were collected for further qualitative analysis. Due to the large number of samples, the shaft speed variable was excluded from the oil quality analysis. 

### 2.2. Physicochemical Characteristics of Hemp Oil

Based on observations, hemp oils, regardless of the variety, have a dark green color with high intensity. The moisture content of the grains was similar, but it was verified whether other factors caused an increase of water content in the oil. The results are presented in Table 2.

Color analysis showed no significant difference in the clarity of ‘Finola’ and ‘Earlina’ oil. It was found that the color intensity was higher in ‘S. Jubileu’, while the values did not matter. In ‘S. Jubileu’, the color of oil was significantly different. When ‘S. Jubileu’ and ‘Finola’ were cold-pressed, the oil was characterized by a lighter green color than the color of the beetles, which was due to a lower amount of chlorophyll content in the oil. Water content was strongly positively correlated with seed size (Table 7), correlation coefficient Cc = 0.83. The larger the seeds, the higher the water content in the oil.

### 2.3. Fatty Acid Profile of Hemp Oil

Statistical analysis showed that the fatty acid profile (FA) of hemp oils depended significantly (*p* < 0.05) on the variety of seeds from which they were obtained. However, the effect of the pressing temperature used and the interaction of the two independent variables included in the MANOVA were non-significant (Table 4). Therefore, a post hoc statistical analysis procedure (Duncan’s test) was performed to compare means only in relation to the varietal factor (‘Finola’, ‘Earlina’, S. Jubileu; Table 3).

**Table 3 molecules-27-03171-t003:** Fatty acids profile [%] and lipid quality indicators in oils obtained from different hemp varieties.

Hemp Seeds Oils									
	‘Finola’ (C)	‘Finola’ (H)	‘Finola’ (mean)	‘Earlina’ (C)	‘Earlina’ (H)	‘Earlina’ (mean)	‘S. Jubileu’ (C)	‘S. Jubileu’ (H)	‘S. Jubileu’ (mean)
C16:0	6.52 ± 0.01	6.61 ± 0.02	6.57 ± 0.06 a	6.25 ± 0.01	6.30 ± 0.01	6.27 ± 0.03 b	6.27 ± 0.00	6.28 ± 0.02	6.28 ± 0.01 b
C16:1 n-7	0.07 ± 0.01	0.07 ± 0.01	0.07 ± 0.00 b	0.08 ± 0.00	0.09 ± 0.01	0.08 ± 0.01 a	0.06 ± 0.02	0.05 ± 0.00	0.05 ± 0.01 b
C17:0	0.02 ± 0.00	0.01 ± 0.00	0.02 ± 0.00 a	nd	nd	nd	nd	nd	nd
C18:0	3.20 ± 0.01	3.19 ± 0.01	3.20 ± 0.01 b	3.10 ± 0.02	3.21 ± 0.02	3.16 ± 0.06 b	3.53 ± 0.03	3.60 ± 0.05	3.57 ± 0.05 a
C18:1 n-9	10.67 ± 0.01	10.56 ± 0.05	10.62 ± 0.07 b	10.34 ± 0.00	10.82 ± 0.01	10.58 ± 0.27 b	11.49 ± 0.13	11.62 ± 0.00	11.64 ± 0.15 a
C18:1 n-7	0.93 ± 0.00	1.03 ± 0.05	0.98 ± 0.06 a	1.02 ± 0.01	0.77 ± 0.16	0.90 ± 0.17 a	0.89 ± 0.18	0.67 ± 0.06	0.78 ± 0.16 a
C18:2 n-6	54.70 ± 0.01	54.66 ± 0.04	54.68 ± 0.03 a	54.86 ± 0.07	54.86 ± 0.04	54.86 ± 0.05 a	54.56 ± 0.09	54.67 ± 0.15	54.67 ± 0.16 a
C18:3 n-6	4.054 ± 0.02	4.25 ± 0.08	4.15 ± 0.14 a	3.25 ± 0.05	3.61 ± 0.01	3.43 ± 0.21 b	3.62 ± 0.12 b	3.77 ± 0.23 b	3.69 ± 0.17 b
C18:3 n-3	16.02 ± 0.01	16.03 ± 0.01	16.02 ± 0.02 c	18.29 ± 0.08	17.81 ± 0.01	18.05 ± 0.28 a	17.25 ± 0.01	17.26 ± 0.04	17.26 ± 0.02 b
C20:0	1.26 ± 0.00	1.07 ± 0.09	1.17± 0.13 a	0.70 ± 0.03	0.43 ± 0.00	0.56 ± 0.16 b	0.45 ± 0.02	0.34 ± 0.13	0.38 ± 0.09 b
C20:1 n-9	0.54 ± 0.00	0.52 ± 0.01	0.53 ± 0.01 a	0.47 ± 0.00	0.49 ± 0.00	0.48 ± 0.01 b	0.46 ± 0.01	0.48 ± 0.02	0.47 ± 0.01 b
C20:2 n-9	0.07 ± 0.00	0.07 ± 0.01	0.07 ± 0.01 a	0.05 ± 0.00	0.05 ± 0.01	0.05 ± 0.01 b	nd	nd	-
C21:0	1.14 ± 0.00	1.13 ± 0.01	1.14 ± 0.01 a	1.00 ± 0.00	0.93 ± 0.03	0.96 ± 0.04 b	0.90 ± 0.00	0.90 ± 0.00	0.90 ± 0.00 c
C22:0	0.56 ± 0.03	0.57± 0.01	0.57 ± 0.01 a	0.40 ± 0.05	0.42 ± 0.01	0.41 ± 0.03 b	0.40 ± 0.05	0.37 ± 0.04	0.39 ± 0.04 b
C24:0	0.25 ± 0.02	0.23 ± 0.03	0.24 ± 0.01 a	0.19 ± 0.01	0.21 ± 0.00	0.20 ± 0.02 a	0.11 ± 0.07	0.00 ± 0.00	0.05 ± 0.07 b
SFA [%]	12.95 ± 0.04	12.81 ± 0.10	12.88 ± 0.01 a	11.64 ± 0.11	11.50 ± 0.06	11.57 ± 0.10 b	11.66 ± 0.01	11.48 ± 0.15	11.57 ± 0.14 b
UFA [%]	87.05 ± 0.06	87.19 ± 0.11	87.12 ± 0.10 b	88.36 ± 0.11	88.50 ± 0.12	88.43 ± 0.10 a	88.34 ± 0.15	88.52 ± 0.08	88.43 ± 0.24 a
MUFA [%]	12.21 ± 0.01	12.18 ± 0.02	12.20 ± 0.02 b	11.91 ± 0.01	12.17 ± 0.16	12.04 ± 0.18 c	12.89 ± 0.07	12.82 ± 0.04	12.86 ± 0.05 a
PUFA [%]	74.84 ± 0.04	75.01 ± 0.13	74.93 ± 0.12 c	76.45 ± 0.10	76.26 ± 0.05	76.39 ± 0.09 a	75.44 ± 0.22	75.70 ± 0.04	75.57 ± 0.24 b
PUFA/SFA ratio	5.78 ± 0.02	5.86 ± 0.05	5.82 ± 0.06 b	6.57 ± 0.07	6.63 ± 0.03	6.60 ± 0.06 a	6.47 ± 0.02	6.59 ± 0.09	6.53 ± 0.10 a
n-6/n-3 ratio	3.67 ± 0.00	3.68 ± 0.01	3.67 ± 0.01 a	3.18 ± 0.01	3.28 ± 0.00	3.23 ± 0.06 c	3.37 ± 0.01	3.38 ± 0.01	3.38 ± 0.01 b
AI	0.07 ± 0.00	0.08 ± 0.00	0.08 ± 0.00 a	0.07 ± 0.00	0.07 ± 0.00	0.07 ± 0.00 b	0.07 ± 0.00	0.07 ± 0.00	0.07 ± 0.00 b
h/H	13.25 ± 0.00	13.09 ± 0.05	13.17 ± 0.05 b	14.04 ± 0.01	13.94 ± 0.03	13.99 ± 0.06 a	14.00 ± 0.03	14.01 ± 0.06	14.01 ± 0.05 a
TI	0.12 ± 0.00	0.12 ± 0.00	0.12 ± 0.00 a	0.10 ± 0.00	0.11 ± 0.00	0.11 ± 0.00 c	0.11 ± 0.00	0.11 ± 0.00	0.11 ± 0.00 b

nd—not detected; C—cold pressed seeds; H—hot pressed seeds; AI—Atherogenicity Index; h/H—Hypocholesterolemic/Hypercholesterolemic Index; TI—Thrombogenicity Index; a, b, c—mean values within a row with different letters are significantly different at *p* < 0.05; values are means of three determinations ± SD.

**Table 4 molecules-27-03171-t004:** The results of the MANOVA multivariate test of fatty acids (method: Wilks test).

Fatty Acids	Value	F	df Effect	df Error	P
Free parameter	0.000000	403,402.8	6	1	0.001205
Variety	0.000001	200.5	12	2	0.004972
Temperature	0.015976	10.3	6	1	0.234481
Variety*Temperature	0.001182	4.7	12	2	0.189336

The nutritional value of the tested oils, based on both the FA profile and the calculated lipid quality indices, was high. Hemp oils are valued for the abundance of unsaturated acids, especially from the PUFA n-6 and PUFA n-3 families. Our study shows that, the main fatty acid of hemp oils was linoleic acid (C18:2 n-6). Its share in the total FA profile exceeded 54% in all analyzed samples (*p* > 0.05). Belonging to the same FA family, γ- linolenic acid (C18:3 n-6) accounted for 3.43 to 4.14% of FA (*p* < 0.05). In turn, the proportion of α-linolenic acid (C18:3 n-3) ranged from 16.02% to 18.05% (*p* < 0.05). It should be stressed that oils obtained from seeds of niche cultivars of Cannabis sativa, i.e., ‘Earlina’ and ‘S. Jubileu’, had a significantly higher total PUFA content than the oil from ‘Finola’ variety, which is popular in Europe and Canada. At the same time, significantly less saturated fatty acids (SFA; average 11.57%) were determined in their FA profile, including palmitic acid (C16:0; 6.28%), arachidonic acid (C20:0; 0.56 and 0.38%, respectively) or heneicosanoic acid (C21:0; 0.96 and 0.90%, respectively). In addition, oil from the ‘S. Jubileu’ cultivar was characterized by the highest proportion of monounsaturated acids (MUFA) (12.86%), with oleic acid (C18:1 n-9; 11.64%) dominating in this group.

The FA profile of hemp oils determined in our study confirms the studies of other authors, although slight differences were observed in the shares of individual compounds. For example, Kiralan et al. [26] determined SFA content from 9.4 to 10.6%, UFA from 89.5 to 90.6%, and PUFA from 73.2 to 78.0% in oils from Cannabis sativa seeds. The work of Alonso-Esteban et al. [27] showed that the main fatty acid (>10%) in hemp oil from unhulled seeds were linoleic acid (54.99–57.36%), α-linolenic acid (12.85–15.87%), and oleic acid (11.92–17.31%). Indeed, the proportion of fatty acids in hemp seeds depends on the variety and time of seed harvest [28] and environmental factors [29], including climatic, agricultural, and light conditions [30].

The nutritional value of oils is determined not only by the profile of the fatty acids they contain, but also by their mutual proportions, which are the basis for calculating so-called lipid quality indices. Hemp seed oils of the varieties ‘Finola’, ‘Earlina’ and ‘S. Jubileu’ were compared in terms of n-6/n-3 ratio, PUFA/SFA ratio, atherogenicity index (AI), thrombogenicity index (TI) and hypocholesterolemic/hypercholesterolemic ratio (h/H) (Table 4).

The products of n-3 and n-6 PUFA metabolism affect cellular biochemical processes, but their different structure determines their other activity. Therefore, a balanced amount of both fatty acid groups in the diet is associated with beneficial health effects. According to EFSA [31], the optimal ratio of FA n-6 to n-3 should be 4:1. However, in the diet of Western societies, a high intake of n-6 fatty acids is observed with an insufficient intake of n-3 fatty acids (15:1 on average) [32]. This imbalance carries an increased risk of NAFLD (non-alcoholic fatty liver disease) [33], cardiovascular disease [34], and obesity [35], among others. Considering the current recommendations for PUFA intake and the important role of these compounds in human nutrition, hemp oils can be considered as products with very promising health-promoting potential. Our study clearly shows that they are characterized by a favorable n-6/n-3 ratio, not exceeding 4:1. Especially valuable in this respect was the oil pressed from seeds of the ‘Earlina’ variety (3.23:1). Similar values of the discussed lipid health quality index for hemp oils were reported by other authors [36,37]. However, Callaway and Pate [38] pointed out that the high content of polyunsaturated fatty acids in hemp oils results in their increased susceptibility to lipid oxidation, which negatively affects the quality and storage time of the oils. The stability of these products can be increased by a multi-stage bleaching process. This treatment removes chlorophylls, which are abundant in hemp oils. Indeed, the mentioned pigments are considered an essential factor in PUFA peroxidation [39].

PUFA/SFA ratio is usually used to assess the effect of diet on cardiovascular function. It is based on the hypothesis that all PUFAs in the diet have hypocholesterolemic properties, whereas SFAs contribute to high serum cholesterol levels. Consequently, the higher the ratio, the more positive the health-promoting effect [40]. It has been shown that oils pressed from ‘Earlina’ and ‘S. Jubileu’ hemp seeds had a higher PUFA to SFA ratio (>6.50; *p* > 0.05) than that determined in ‘Finola’ seed oil and a more favorable h/H ratio (about 14; *p* > 0.05). This ratio determines the relationship between hypo- and hypercholesterolemic FA [41], complementing the PUFA/SFA relationship. Lower values of the h/H ratio indicate a less beneficial effect on the cardiovascular system. Atherogenicity (AI) and thrombogenicity (TI) indices were also determined for the tested oil samples. In both cases, these values were significantly higher for the oil pressed from seeds of the ‘Finola’ variety (0.08 and 0.12, respectively).

Hemp oils, regardless of their varietal differentiation in terms of FA profile, are therefore characterized by more favorable lipid quality indices compared to popular, commonly used edible oils such as palm oil, soybean oil or olive oil [42]. According to Szabo et al. [42], the AI index values in these fresh oils are 0.895; 0.128; 0.17, and the values of the TI index 1.830; 0.247; 0.399, respectively. Moreover, the PUFA/SFA values determined in ‘Finola’, ‘Earline’ and ‘S. Jubileu’ oils in our study were much higher than those determined by Kim et al. [43] in rapeseed oil (4.63), corn oil (4.25), soybean oil (4.16), sesame oil (3.21), and olive oil (0.56).

### 2.4. Identification of Sterols in Hemp Oil

PUFAs present in hemp seed oils are valuable sources of compounds desirable in the human diet. Part of the unsaponifiable fraction of hemp seed oil is phytosterols. Phytosterols block the absorption of cholesterol, thus reducing hypercholesterolemia. They also have antiviral, antifungal, and anti-inflammatory properties [44]. Phytosterols also enhance the antioxidant activity of the oil [45]. In the present study, irrespective of hemp variety and oil extraction method, the content of β-sitosterol was the highest (1.23 to 1.35 mg/g-Table 5). Additionally, in the study of Monserrat-de la Paz et al. [10], among the phytosterols, β-sitosterol was the highest amount, and it was a similar amount (1.91 mg/g). Campesterol, Δ5-avenasterol, Δ7-avenasterol, Δ5,24-stigmastadienol, stigmasterol, campestanol, and 24-methylenecycloartenol were also found in the hemp seed oil samples tested in decreasing amounts. Siano et al. [44] showed only the presence of β-sitosterol, campesterol, Δ5-avenasterol, and stigmasterol. The hemp seed oils studied by Velickovska et al. [46] had the most β-sitosterol, campesterol, Δ5-avenasterol, stigmasterol, Δ7-stigmasterol, sitostanol, and campestanol.

The statistical MANOVA analysis, showed no significant difference in sterol content depending on the type of oil and the temperature of the seeds used for pressing (Table 6). In the correlation analysis, no strong relationship was found between the factors and the independent variables. The highest phytosterols content were in the popular cold-pressed variety ‘Finola’ (2.13 mg/g) and the hot-pressed variety (2.06 mg/g). The least phytosterols in cold-pressed oil from the variety ‘Earlina’ (1.95 mg/g). In various studies on phytosterol determinations in hemp oil, total phytosterols ranged between 2.19 mg/g and 6.7 mg/g [10,17,25,47,48] This means that the samples studied had values close to the lowest among the results of hemp oils available in the literature. It is recommended to consume 2–3 g of phytosterols/stanols per day, so hemp oil can be one of the dietary elements that will provide these components [49].

### 2.5. Principal Components Analysis

Principal component analysis (PCA) was applied to observe possible clusters in the oil obtained from three different hemp seeds. In all PCA loading plots (Figure 1) the first two principal factors accounted for 82.5% (Dim1 = 50.9% and Dim2 = 31.6%) of the total variation. The PCA results showed noticeable differences between the hemp oils. Dimension 1 was mainly positively correlated with saturated fatty acid share (SFA) (r = 0.955), total sterols content (r = 0.931), ω6/ω3 fatty acid ratio (r = 0.871) and Thrombogenicity Index (TI) (r = 0.846). It was also negatively correlated with unsaturated fatty acid (UFA) (r = −0.965), hypocholesterolemic/Hypercholesterolemic Index (h/H) (r = −0.951), and PUFA/SFA ratio (r = −0.941). Dimension 2 was mainly positively correlated with the L (r = 0.808) and b (r = 0.806) parameter of Lab color. It was also negatively correlated with the water content (r = −0.950), and a parameter (r = −0.922) of Lab color. The data presented in the score plot divide the samples into three groups related to hemp variety and show a slight influence of the temperature used for the pressing process. However, in the case of the ‘Finola’ seed variety, we observe a large distance between the samples obtained during pressing at low (C) and high (H) temperatures.

The analysis of the data, suggests that the water content has the greatest impact on the differentiation of the samples. In addition, the samples on the right side of the Y-axis (‘Finola’) have the highest phytosterols content in the oil (from 2.06 to 2.13 mg/g oil), including β-sitosterol and saturated fatty acids (SFA), which are 12.94 and 12.79% in oil pressed at low and high temperatures, respectively. On the left side of the Y-axis, two clusters separate the oil samples obtained from the ‘Earlina’ and ‘S. Jubileu’ varieties. Above the X-axis are oils samples obtained from ‘Earlina’ variety, which are characterized by the lowest water content in the oil (121 and 127 ppm in oils obtained at low and high temperatures, respectively), and a higher intensity of the yellow color. Below the X-axis, are the oil samples of the Jubileu variety, characterized by the highest water content in the oil (153 and 159 ppm in oils obtained at low and high temperatures, respectively), and a higher intensity of green color.

## 3. Materials and Methods

### 3.1. Materials

Seeds of hemp seed from an approved crop were used for this study. The seeds were harvested in September 2021 from a Polish plantation. Seeds were stored in a dry and cool place. After harvesting, the hemp seeds were dried with air (30–35 °C) to the humidity of approx. 8–11% and stored in a dry and airy room with humidity <10%. Notch contamination <1%. Three *Cannabis sativa* cultivars were used for this study: ‘Finola’ (FIN-314), ‘Earlina8FC’, and Secuieni Jubileu. Industrial seeds, commercially used for oil production, were used. ‘Earlina 8FC’ and ‘S. Jubileu’ are a little recognized variety among 600 Cannabis cultivars [50,51]. They are used industrially for vegetable oil production; however, there is little information about these varieties in the literature. *Cannabis sativa* ‘Finola’ (previously “FIN-314”) is a dioecious cultivar characterized by early sowing and flowering, low branching and short length, with smaller seeds and lower levels of biomass recovery than other hemp varieties; ‘Finola’ was first accepted in Finland in the list of official plant cultivars, in 2003 and was published in the EU Common Catalogue. ‘Finola’, as well as the other hemp cultivars, is a low-maintenance crop which needs no herbicides or other pesticides [52]. 

### 3.2. Pressing Process

The aim of the study was to compare the quality of the obtained oil from three hemp varieties concerning changes in the technological parameters during extraction by single-stage pressing. The study consisted of three stages: analysis of raw material, measurement of technical parameters of pressing, and quality analysis of obtained oils. The data set was subjected to statistical analysis to verify the influence of technical and technological parameters on the quality and quantity of obtained oil after pressing.

The seeds were subjected to granulometric analysis by sieve method, bulk density by weight method in a 1 dm^3^ measuring cylinder, moisture content by means of a weighing machine at 120 °C and fat content (Soxhlet extraction). Each measurement was performed several times for different seed portions. The results are shown in Table 7.

From the collected data in Table 7, it can be seen that there are no significant differences in the parameters of the compared seeds.

In the next step, the seeds were subjected to the pressing process using a screw press with variable shaft speed parameters. The seed temperature was 20 °C and 60 °C and the shaft speed was 33 rpm and 70 rpm. For each case, 4 pressing trials of a preset mass of seeds were performed. The material was not pre-crushed and hulled. The oil obtained was filtered on Buchner funnels under vacuum through a cellulose membrane. Figure 2 shows the oil pressing and filtration station. 

In the next stage of the study, the evaluation of the effect of rotational speed on oil quality was abandoned. Based on our own experience and the literature, no impact of technical parameters of the pressing process on oil quality was found. However, the effect of seed pressing temperature was found, and this variable was taken into account in further studies. In the third stage of the study, the fatty acid profile and the proportion of sterols were determined by the GC method, as well as, the oil color by the Lab method using chroma meter Cr-400 and the proportion of water by the Karl Fischer method using Aquamax KF Coulometric.

### 3.3. Determination of the Fatty Acid Profile Using GC-FID

Hemp oils were saponified with 0.5 M KOH in methanol. Transesterification of fatty acids with the BF_3_ (boron trifluoride) solution in methanol was carried out using official AOCs Ce 2–66 method [53].

The conditions for chromatographic separation conformed with the procedure described by Wołoszyn et al. [54]. The methyl esters of fatty acids were quantified by GC-FID (Agilent 7890 A series, Agilent Tech. Inc., St. Clara, CA, USA) using a J&W Scientific HP-88 series 100 m × 0.25 mm × 0.20 μm fused silica capillary column (Agilent Tech. Inc., St. Clara, CA, USA) and a flame-ionization detector (FID) from Agilent Tech.

### 3.4. Lipid Health Indicators

According to the fatty acid profile analysis, selected nutritional quality parameters of hemp oils were calculated:
(1)PUFA/SFA ratio = (ΣDiUFA + ΣTriUFA + ΣTetraUFA)/ΣSFA [41];(2)n-6/n-3 PUFA ratio = (C18:2 n-6 + C18:3 n-6)/(C18:3 n-3 + C18:4 n-3) [41];(3)AI (Atherogenicity Index) = (C12:0 + 4 × C14:0 + C16:0)/ΣUFA [41];(4)TI (Thrombogenicity Index) = (C14:0 + C16:0 + C18:0)/[(0.5 × MUFA) + (0.5 × Σn-6) + (3 × Σn-3) + (Σn-3/Σn-6)] [55];(5)h/H (Hypocholesterolemic/Hypercholesterolemic Index) = [(C18:1 n-9 + C18:1 n-7 + C18:2 n-6 + C18:3 n-6 + C18:3 n-3 + C20:3 n-6 + C20:4 n-6 + C20:5 n-3 + C22:4 n-6 + C22:5 n-3 + C22:6 n-3)/(C14:0 + C16:0)] [56].

### 3.5. Method for Determining Sterols

For analysis of sterols, a total of 0.05 g of each hemp oil was used. To the samples were added 50 µg of internal standard(5α-cholestane-Supelco, Bellefonte, PA, USA). The samples were saponified with 1 M KOH in methanol, and the unsaponifiables were extracted using a mixture of hexane and methyl tert-butyl ether (1:1, *v/v*). The solvent was evaporated under a nitrogen stream, and dry residues were dissolved in anhydrous pyridine (Supelco, Bellefonte, PA, USA), and silylated with BSTFA + 1% TMCS (Supelco, Bellefonte, PA, USA). The phytosterols were analyzed using a Hewlett-Packard 6890 gas chromatograph (Agilent Technologies, Palo Alto, CA, USA) in splitless mode with an FID detector and a DB-35MS capillary column (25 m × 0.20 mm, 0.33 μm; Agilent J&W, USA). The detector and injector were set at a temperature of 300 °C. The oven temperature was initially 100 °C for 5 min, increasing at 25 °C/min to 250 °C and then at 3 °C/min to 290 °C. The final temperature was held for 20 min. The carrier gas was hydrogen and the flow rate was 1.5 mL/min. Sterols were identified by comparing their retention times with those of standards. The sterols were determined in duplicate [57].

## 4. Conclusions

Cold-pressed, unrefined vegetable oils are a valuable source of energy and vitamins for humans. Whether conventional pressing of little-known hemp seed varieties Earlin ‘Earlin 8FC’ and Secuieni Jubileu affects their quality was investigated. For comparison, the well-known variety Finola ‘Finola’ was used as a reference. The seeds differed in size and oiliness, while oil extraction from them by pressing is highly efficient. The highest pressing efficiency (84% on average, irrespective of pressing parameters) was obtained with ‘S. Jubileu’, whose average diameter was significantly larger than the other cultivars. A dark color on average L* = 23.0 characterized the oil in all cases, with a green-yellow hue a* (1.33–2.04) and b* (2.84–5.47). A strong correlation was also observed between the oil’s water content and the seeds’ size, whose moisture content was comparable. The amount of water in oil was trace and did not exceed 160 ppm. The oils of cultivar ‘Earlina’ and ‘S. Jubileu’ has higher UFA content (average 88%) than ‘Finola’ (87%). All oil samples contained comparable sterols, with campesterol (0.32 mg/g), β-sitosterol (1.3 mg/g) nad Δ5-avenasterol (0.15 mg/g) predominating. Technological parameters of the pressing process did not affect the profile of fatty acids and the content of sterols in the oil significantly while preheating the seeds to 60 °C caused an improvement in the efficiency of the process by 2.9% on average at slower pressing speeds and by 11% at higher speeds. Increasing the shaft speed increased the pressing efficiency by 5.4%, and 13.5% of seeds preheated to 60 °C. The critical parameters had no significant effect on the quality of the extracted oil, while they significantly affected the yield (P) and the quantity of extracted oil (E). The process’s highest efficiency was obtained for preheated seeds pressing at slow speeds of the press (on average 88.0%) and the lowest for unprocessed seeds at fast speeds of the shaft (on average 71.5%). The organoleptic analysis also revealed the oils from ‘Earlina’ and ‘S. Jubileu’ were devoid of bitter taste compared to ‘Finola oil’.

## Figures and Tables

**Figure 1 molecules-27-03171-f001:**
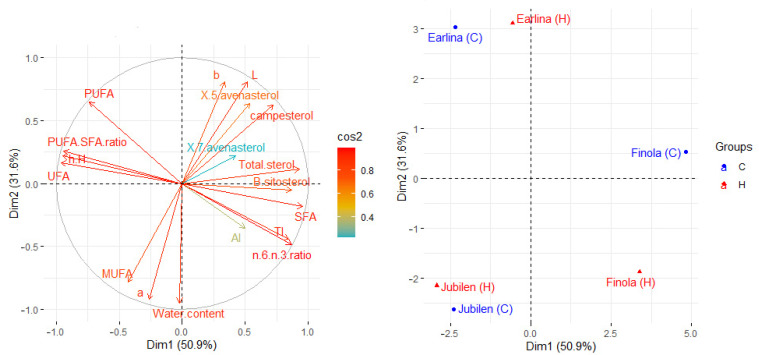
Principal component analysis (PCA) of the loadings plot and the score plot of data from fatty acid groups (SFA, UFA, MUFA, PUFA), PUFA/SFA ratio, n-6/n-3 ratio, Atherogenicity Index (AI), hypocholesterolemic/Hypercholesterolemic Index (h/H), Thrombogenicity Index (TI), phytosterols, color and water content of hemp oils.

**Figure 2 molecules-27-03171-f002:**
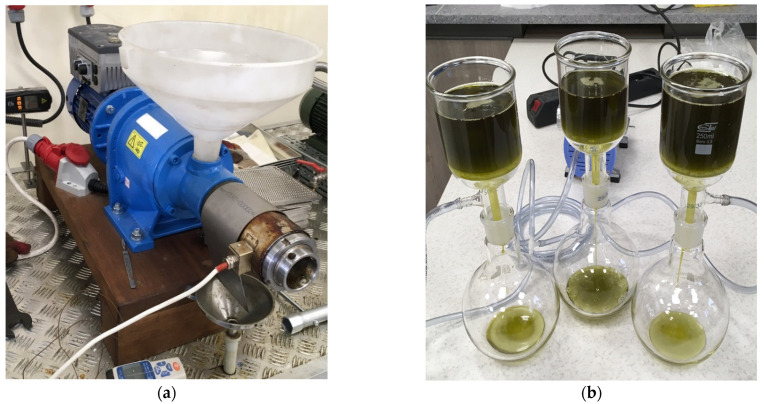
Test stand: (**a**) oil pumping; and (**b**) oil filtering station.

**Table 1 molecules-27-03171-t001:** Performance and efficiency of the hemp seed pressing process under varying process parameters.

Variety	Pressing Parameters *	T [°C]	P [kg/h]	E [%]
**‘Finola’**	33 rpm/20 °C	45.4 ± 0.3	8.43 ± 0.16	82.4 ± 1.9
	70 rpm/60 °C	57.4 ± 0.1	17.31 ± 1.10	81.6 ± 2.4
	33 rpm/60 °C	62.13 ± 0.4	11.69 ± 0.26	86.0 ± 4.0
	70 rpm/20 °C	41.0 ± 0.5	10.47 ± 0.31	67.0 ± 1.6
**‘Earlina 8FC’**	33 rpm/20 °C	44.5 ± 1.2	9.00 ± 0.21	84.6 ± 0.9
	70 rpm/60 °C	52.7 ± 0.5	16.78 ± 1.40	81.5 ± 2.2
	33 rpm/60 °C	55.2 ± 0.3	10.85 ± 0.23	87.2 ± 2.3
	70 rpm/20 °C	42.0 ± 0.7	11.60 ± 0.37	73.9 ± 1.7
**‘S. Jubileu’**	33 rpm/20 °C	49.1 ± 0.2	8.88 ± 0.24	87.9 ± 2.7
	70 rpm/60 °C	57.2 ± 0.4	17.05 ± 1.10	84.5 ± 1.3
	33 rpm/60 °C	60.8 ± 0.5	11.13 ± 0.47	90.6 ± 1.7
	70 rpm/20 °C	46.4 ± 0.12	11.51 ± 0.19	73.6 ± 1.6

* 33/70 rpm-press shaft speed; 20/60 °C–hemp seed temperature. Values are means of three determinations ± SD. T—temperature of hemp seed, P—press performance, E—process efficiency.

**Table 2 molecules-27-03171-t002:** Results of color and water content in hemp oil.

Parametry	‘Finola’ (C)	‘Finola’ (H)	‘Earlina’ (C)	‘Earlina’ (H)	‘S. Jubileu’ (C)	‘S. Jubileu’ (H)
**L***	23.31 ± 0.01	23.22 ± 0.01	23.42 ± 0.01	23.59 ± 0.01	22.03 ± 0.01	22.06 ± 0.01
**a***	1.33 ± 0.01	1.89 ± 0.01	1.43 ± 0.01	1.12 ± 0.01	2.04 ± 0.01	1.92 ± 0.01
**b***	5.47 ± 0.01	2.99 ± 0.01	4.94 ± 0.01	4.49 ± 0.01	2.84 ± 0.01	3.21 ± 0.01
**Water content [ppm]**	142 ± 2	146 ± 3	121 ± 2	127 ± 1	153 ± 3	159 ± 2

Values are means of three determinations ± SD; C—cold pressed seeds; H—hot pressed seeds; L*–values for perceptual lightness; a* and b*—values for the four unique colors of human vision: red, green, blue, and yellow.

**Table 5 molecules-27-03171-t005:** Phytosterols content [mg/g] in oils obtained from different hemp varieties and pressing temperature.

Phytotserol [mg/g]	Hemp Seeds Oils					
	‘Finola’ (C)	‘Finola’ (H)	‘Earlina’ (C)	‘Earlina’ (H)	‘S. Jubileu (C)	‘S. Jubileu’ (H)
campesterol	0.33 ± 0.00	0.33 ± 0.01	0.32 ± 0.01	0.33 ± 0.02	0.29 ± 0.02	0.29 ± 0.02
campestanol	0.03 ± 0.00	0.03 ± 0.00	0.02 ± 0.00	0.03 ± 0.00	0.03 ± 0.00	0.03 ± 0.00
stigmasterol	0.05 ± 0.01	0.05 ± 0.01	0.04 ± 0.00	0.04 ± 0.01	0.04 ± 0.00	0.03 ± 0.00
β-sitosterol	1.35 ± 0.02	1.28 ± 0.00	1.23 ± 0.09	1.27 ± 0.09	1.26 ± 0.11	1.25 ± 0.07
sitostanol	0.02 ± 0.00	0.04 ± 0.00	0.02 ± 0.00	0.03 ± 0.00	0.02 ± 0.01	0.02 ± 0.01
Δ5-avenasterol	0.16 ± 0.00	0.15 ± 0.02	0.15 ± 0.01	0.16 ± 0.02	0.14 ± 0.02	0.15 ± 0.02
Δ5,24-stigmastadienol	0.06 ± 0.00	0.06 ± 0.01	0.05 ± 0.00	0.06 ± 0.01	0.04 ± 0.01	0.05 ± 0.00
Δ7-avenasterol	0.10 ± 0.00	0.08 ± 0.00	0.08 ± 0.00	0.09 ± 0.01	0.10 ± 0.01	0.05 ± 0.00
24-methylenecycloartenol	0.03 ± 0.00	0.03 ± 0.00	0.03 ± 0.01	0.04 ± 0.00	0.04 ± 0.00	0.04 ± 0.01
TOTAL	2.13 ± 0.03	2.06 ± 0.03	1.95 ± 0.10	2.05 ± 0.15	1.97 ± 0.20	1.97 ± 0.11

C–cold pressed seeds; H–hot pressed seeds; values are means of two determinations ± SD.

**Table 6 molecules-27-03171-t006:** The results of the MANOVA multivariate test of phytosterols (method: Wilks test).

Phytosterols	Value	F	df Effect	df Error	P
Free parameter	0.000163	1020.114	6	1	0.023962
Variety	0.002106	3.465	12	2	0.245647
Temperature	0.098883	1.519	6	1	0.551892
Variety*Temperature	0.014996	1.194	12	2	0.543322

**Table 7 molecules-27-03171-t007:** Parameters of hemp seed.

Variety	Fat Content [%]	Average Seed Size [mm]	Seed Density [kg/m^3^]	Moisture [%]
**‘Finola’**	30.45 ± 0.85	1.152 ± 0.06	516.06 ± 3.89	8.39 ± 0.09
**‘Earlina 8FC’**	29.47 ± 0.84	1.096 ± 0.02	521.00 ± 1.55	8.32 ± 0.09
**‘S. Jubileu’**	31.03 ± 1.03	1.213 ± 0.01	506.73 ± 1.76	8.31 ± 0.10

## Data Availability

Not applicable.

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
