# Peer review of "Quality of Oil Pressed from Hemp Seed Varieties: ‘Earlina 8FC’, ‘Secuieni Jubileu’ and ‘Finola’"

_molecules, 2022, doi:10.3390/molecules27103171_

Round 1

Reviewer 1 Report

Dear authors, 

This is important submission towards Molecules. The results are very interesting and the manuscript worth to be considered at Molecules. However, I see some major revisions should be incorporated to improve the submission. My comments and suggestions are included in enclosed file.

Please explain why your seed oil yield values are so high as compared to those reported in literature ?

Please note that English must be improved.

Regards.  

Author Response

This is important submission towards Molecules. The results are very interesting and the manuscript worth to be considering Molecules. However, I see some major revisions should be incorporated to improve the submission. My comments and suggestions are included in enclosed file.

Thank you for your comments in the appendix. We have made significant changes to the article to improve the readability and understanding of the research results presented.

In table 3, we have carefully analysed all values. The corrections have been taken into account, and now the sum of FA in all samples is 100%. In the previous version, the differences in the sum of these compounds were insignificant, ranging from 0.03-0.07%. This was due to the accuracy of the results (automatic + manual integration of the peaks, calculations in Excel, rounding the results to two significant numbers). At the same time, we would like to inform you that minor adjustments in the table did not affect the statistical interpretation of the results.

Please explain why your seed oil yield values are so high as compared to those reported in literature ?

The efficiency of the process of extracting oil from seeds is variously interpreted. The most common is the ratio of oil weight to grain weight. Of course, the values will be much lower. We have proposed the efficiency of the process, i.e. the mass of oil after pressing to the mass of fat in the seeds. This method was verified on rapeseed, and we also obtained similar values of 70-90%, which means that 10-30% of the fat remains in the pomace. The pressing of hemp was done with several repetitions for each variant of parameters. For the study, we used 300 kg of hemp seeds. The efficiency and effectiveness of pressing were measured only after heating the press, i.e. after ten min. from turning it on. We poured 5 kg of seeds and measured the time. On the basis of the obtained masses, we calculated the efficiency of the process and performance. Therefore, it was important in the methodology to cite the relationship from which the efficiency was calculated.

Please note that English must be improved.

The text has been corrected for English.

Reviewer 2 Report

(1)The title of the article is about the identification of sterols in the pressed oils of different hemp seeds, but the research content of the article is a bit confusing, and the "Identification of sterols in hemp oil" section of the article only focuses on different unsaturated fatty acids in different seed oils,health effects of unsaturated fatty acid composition and proportions; no information on how the pressing process affects the total sterol content. It is suggested to change the title and reorganize the frame and content of the article. The abstract section describes most of the pressing and physicochemical characteristics, however what does the content of total sterols in this paper indicate? Nevertheless the final conclusion is not too appropriate to the research content of the topic and needs to be revised.

(2)Please unify the expression of "Mo" on page 2 of the article, whether it is "the weight of seeds" or "the weight of the oil", and write the correct formula: "The efficiency of the press"(P)and "The process efficiency"(E).

(3)What does Table 1 illustrate? (The word in Table 1 is spelled wrong in the header).  If it is to optimize the hemp oil pressing process, first, the number of parameter comparison groups is not enough (only 2 variables for one parameter are not enough); and second, what are the final optimized process conditions, and how many about "the efficiency of the press"(P)and "the process efficiency"(E)under the optimized conditions? Furthermore,the next research should take the oil obtained under this optimized condition as the research object.

(4)What about do L*, a*, b* in Table 2 mean?

(5)It is suggested that Δ5-avenasterol is also included in the statistics of total sterol content in this article.

Author Response

1)The title of the article is about the identification of sterols in the pressed oils of different hemp seeds, but the research content of the article is a bit confusing, and the "Identification of sterols in hemp oil" section of the article only focuses on different unsaturated fatty acids in different seed oils,health effects of unsaturated fatty acid composition and proportions; no information on how the pressing process affects the total sterol content. It is suggested to change the title and reorganize the frame and content of the article. The abstract section describes most of the pressing and physicochemical characteristics, however what does the content of total sterols in this paper indicate? Nevertheless the final conclusion is not too appropriate to the research content of the topic and needs to be revised.

We fully agree with this comment. The assumptions for the article were different from the final form. The changes made before submitting the article resulted in poor readability of the whole article. We have made significant changes to the text. Now the title, aim and conclusions correspond with the subject of the study.

(2)Please unify the expression of "Mo" on page 2 of the article, whether it is "the weight of seeds" or "the weight of the oil", and write the correct formula: "The efficiency of the press"(P)and "The process efficiency"(E).

We have clarified the description of the relationships when counting the process efficiency.

(3)What does Table 1 illustrate? (The word in Table 1 is spelt wrong in the header).  If it is to optimize the hemp oil pressing process, first, the number of parameter comparison groups is not enough (only 2 variables for one parameter are not enough); and second, what are the final optimized process conditions, and how many about "the efficiency of the press"(P)and "the process efficiency"(E)under the optimized conditions? Furthermore, the next research should take the oil obtained under this optimized condition as the research object.

I fully agree that 4 measuring points of the process do not make it possible to perform process characterization and, of course, optimization calculations. The purpose of our study was only a preliminary analysis of how changes in the pressing parameters affect the quality of oil and the quantity of oil obtained. In addition, we wanted to describe two varieties of hemp, Earlin and S. Jubileum, which are little known in the literature. The determination of efficiency (E) is derived from the ratio of the amount of oil pressed to the oil contained in the seeds. This is how process efficiency is determined for different types of technological machines, i.e. actual value to theoretical value (in our case the fat contained in the seeds).

I agree that it is worth doing an optimization study, but it would also need to take into account geometric changes in the press, which is difficult for us to do because we are not in the mechanical engineering business. Our results confirm that it is possible to increase the oil yield without affecting its quality. I hope that they will be an inspiration for teams from technical universities. 

(4)What about do L*, a*, b* in Table 2 mean?

These are the parameters that determine the colour of products such as food. L* colour intensity, 0 - very intense and 100 - colourless. a* = 100 red, -100 green, b* =100 yellow and -100 blue.  Measurement with Chroma Meters for determining the colour quality of liquid and solid substances. Commonly used to evaluate food products. This article represents the beginning of our research, the next work will be about the oil bleaching process of the oil samples presented in this article.

(5)It is suggested that Δ5-avenasterol is also included in the statistics of total sterol content in this article.

From the PCA analysis, the Δ5-avenasterol content of hemp oils was not differentiated by effect factors. Values for DIM 1 r=0.52 were obtained and for DIM 2 r=0.64 only values >0.8 were reported as differentiating factors. 

Round 2

Reviewer 1 Report

Dear Authors, 

Several points raised in the first revision were not taken into account and the manuscript still has some weaknesses and flaws that should be addressed to be considered at Molecules with its associated impact factor and readership. 

1/Title is about two cultivars, however, in the entire manuscript there are three cultivars;

2/State-of-the-art of research must be improved (please my comment in the attached file), 

3/English should be improved. 

Regards.  

Author Response

Thank you for once again highlighting the suggestions you sent. The introduction has been updated with the literature provided. I agree that hemp is primarily known for its psychoactive properties and is little known for its phytochemicals. The indicated lithium sources are a valuable addition to the literature review.
The presented research was mainly focused on the recognition of 'Earlina' and 'S. Jubileu' cultivars and 'Finola' were studied as a reference sample. After discussion with the manuscript's authors, we decided that all the varieties studied should be indicated in the title. 
We re-examined the quality of the English and inserted changes as much as we could improve. 
We would like to thank you for such an insightful review, and we do not doubt that the comments you sent us were valuable and increased the value of our article. 

Reviewer 2 Report

The author has carefully answered and revised the 5 questions in the first reviewers' comments, and has also revised the relevance between the title and the content of the article, abstract and conclusion of the article. Therefore, it is recommended to accept.

Author Response

Thank you for pointing out significant flaws in the manuscript. The corrections made have also, in our opinion, improved the quality of the presented content.